# The species distribution and antimicrobial resistance profiles of *Nocardia* species in China: A systematic review and meta-analysis

**Chaohong Wang[1], Qing Sun[1], Jun Yan[1], Xinlei Liao[1], Sibo Long[1], Maike Zheng[1], Yun Zhang[2], Xinting Yang[2], Guangli Shi[1], Yan Zhao[1], Guirong Wang** [1]*, **Junhua Pan[3]***

**1** Department of Clinical Laboratory, Beijing Chest Hospital, Capital Medical University, Beijing Tuberculosis and Thoracic Tumor Institute, Beijing, China, **2** Tuberculosis Department, Beijing Chest Hospital, Capital Medical University, Beijing, China, **3** Beijing Chest Hospital, Capital Medical University, Beijing Tuberculosis and Thoracic Tumor Institute, Beijing, China

* wangguirong1230@ccmu.edu.cn (GW); pxm1960@sohu.com (JP)

**Data Availability Statement:** All relevant data are within the paper and its Supporting information files.

## Abstract

### Background

*Nocardia* species can cause local or disseminated infection. Prompt diagnosis and appropriate treatment of nocardiosis are required, because it can cause significant morbidity and mortality. Knowledge of local species distribution and susceptibility patterns is important to appropriate empiric therapy. However, knowledge on the epidemiology and antimicrobial susceptibility profiles of clinical *Nocardia* species remains limited in China.

### Methods

The data of isolation of *Nocardia* species were collected from databases such as Pubmed, Web of Science, Embase as well as Chinese databases (CNKI, Wanfang and VIP). Meta-analysis was performed using RevMan 5.3 software. Random effect models were used and tested with Cochran's Q and $I^2$ statistics taking into account the possibility of heterogeneity between studies.

### Results

In total, 791 *Nocardia* isolates were identified to 19 species levels among all the recruited studies. The most common species were *N. farcinica* (29.1%, 230/791), followed by *N. cyriacigeorgica* (25.3%, 200/791), *N. brasiliensis* (11.8%, 93/791) and *N. otitidiscaviarum* (7.8%, 62/791). *N. farcinica* and *N. cyriacigeorgica* were widely distributed, *N. brasiliensis* mainly prevalent in the south, *N. otitidiscaviarum* mainly distributed in the eastern coastal provinces of China. Totally, 70.4% (223/317) *Nocardia* were cultured from respiratory tract specimens, 16.4% (52/317) from extra-pulmonary specimens, and 13.3% (42/317) from disseminated infection. The proportion of susceptible isolates as follows: linezolid 99.5% (197/198), amikacin 96.0% (190/198), trimethoprim-sulfamethoxazole 92.9% (184/198), imipenem 64.7% (128/198). Susceptibility varied by species of *Nocardia*.

**Funding:** Funding was provided by Capital's Funds for Health Improvement and Research (2022-1G-2162) to JP, Beijing Public Health Experts Project (2022-3-040), Beijing Tongzhou Municipal Science & Technology commission (KJ2022CX044) and Tongzhou Yunhe Project under Grant (YH201917) to GW. The funders had no role in study design, data collection and analysis, decision to publish, or preparation of the manuscript.

**Competing interests:** The authors have declared that no competing interests exist.

## Conclusions

*N. farcinica* and *N. cyriacigeorgica* are the most frequently isolated species, which are widely distributed in China. Pulmonary nocardiosis is the most common type of infection. Trimethoprim-sulfamethoxazole can still be the preferred agent for initial *Nocardia* infection therapy due to the low resistance rate, linezolid and amikacin could be an alternative to treat nocardiosis or a choice in a combination regimen.

### Author summary

Nocardiosis is a neglected tropical disease caused by *Nocardia* and potentially lifethreatening infection. Despite increasing attention towards the *Nocardia* infections, the overall epidemiological information and antimicrobial susceptibility profiles of clinical *Nocardia* species remains limited for China. A systematic review and meta-analysis was conducted using data of 42 qualified publications. Our pooled analysis of these studies demonstrated that *N. farcinica* and *N. cyriacigeorgica* are the most frequently isolated species, which are widely distributed in China. Totally, 70.4% (223/317) *Nocardia* were cultured from respiratory tract specimens. Susceptibility varied by species of *Nocardia*. Trimethoprim-sulfamethoxazole can still be the preferred agent for initial *Nocardia* infection therapy due to the low resistance rate, linezolid and amikacin could be an alternative to treat nocardiosis or a choice in a combination regimen.

## Introduction

The genus *Nocardia* are filamentous, Gram-positive, aerobic, weakly acid-fast bacteria, they are closely related to the genera *Corynebacterium* and *Mycobacterium* [1]. Nocardiosis resembles tuberculosis (TB) and non-tuberculous mycobacteria (NTM) disease in most clinical symptoms and radiological manifestations [2–3], which may lead to misdiagnosis or underdiagnosis. China remains a high TB burden country in 2021 and the prevalence of NTM increased considerably [4]. Although increased awareness of *Nocardia* by clinicians, the knowledge on *Nocardia* infection remains paucity in China.

To date, more than 200 *Nocardia* species have been described (http://www.bacterio.net/genus/nocardia), with different species showing different antibiotic susceptibility patterns [1]. The distribution of *Nocardia* species tends to vary as per geographical regions [5]. Different species may exhibit predilection to certain body sites. A few studies have reported the clinical features, epidemiology and antimicrobial resistance patterns of *Nocardia* species in China [6,7–11], however, these reports harbored high degrees of variability.

The aim of this study was to elucidate the species distribution and antimicrobial resistance profiles of *Nocardia* species in China using meta-analysis based on systematic review of articles published until 30 September, 2022. This study will provide more detailed information to overview the magnitude of *Nocardia* infection and provide guidance on empirical therapy in China.

## Method

### Search strategies

The available literatures were identified by searching in the electronic database such as: Pubmed, Web of Science, Embase as well as Chinese databases (CNKI, Wanfang and VIP),

with medical subject headings (MeSH) terms and a proper use of keywords, published until 30 September, 2022. The search criteria were "*Nocardia*" "nocardiosis" or "*Nocardia* disease" and "China", "Chinese", etc. Both English articles and Chinese articles were considered.

### Inclusion and exclusion criteria

The process of article screening and selection following the Preferred Reporting Items for Systematic Review and Meta-Analyses (PRISMA) 2020 statement guidelines [12]. All original articles which referenced to the standard method for *Nocardia* species identification and/or antimicrobial resistance tests and presented either the cross-sectional or cohort studies from China were included. The standard identification was based on culture, mass spectrometry or molecular methods (e.g. DNA sequencing, multilocus sequence analysis, metagenomic next-generation sequencing). The antimicrobial resistance test was determined using the standard Broth microdilution method, which following the recommendations of the Clinical and Laboratory Standards Institute.

Articles were excluded for any of the following characteristics: (1) reviews, conference presentations, literature reviews, non-full-text and unpublished data; (2) studies with less than 5 cases; (3) isolates were not identified to species level; (4) data from non-Chinese population.

### Data extraction and definitions

After the articles were merged into Excel 2019, the results are de-duplicated and filtered. Two researchers independently extracted the data from eligible studies as follows: first author, year of publication, enrollment time, sample size, province of study, and method of species identification. Inconsistency between the reviewers was resolved through discussion to obtain consensus.

### Statistical analysis

Meta-analysis was performed using RevMan 5.3 software. Stratified analyses were performed with respect to the geographic areas, infected sites, culture methods. Random effect models were used and tested with Cochran's Q and $I^2$ statistics taking into account the possibility of heterogeneity between studies. To assess possible publication bias, value of $P<0.05$ was considered an indication of statistically significant publication bias using the Egger weighted regression methods.

## Results

### Characteristics of the included studies

As shown in Fig 1, a total of 4493 related articles were obtained through literature retrieval of keyword combination in the database. In secondary screening and after duplication, 61 articles were included for detailed full text evaluation. We also aggregated the sample collected time, province and hospital involved in the included articles, and 19 articles were deleted due to duplication. Finally, 42 studies were included in present study [6,7,10,11,13–50], including 28 published in Chinese and 14 in English, covering 20 provinces in mainland China and involving 1008 clinical Nocardia isolates. Table 1 summarizes the characteristics of the selected articles.

### Prevalence of different *Nocardia* species

In total, 791 *Nocardia* isolates were identified to 19 species levels among all the recruited studies (Table 2). The most common species were *N. farcinica* (29.1%, 230/791), followed by *N.*

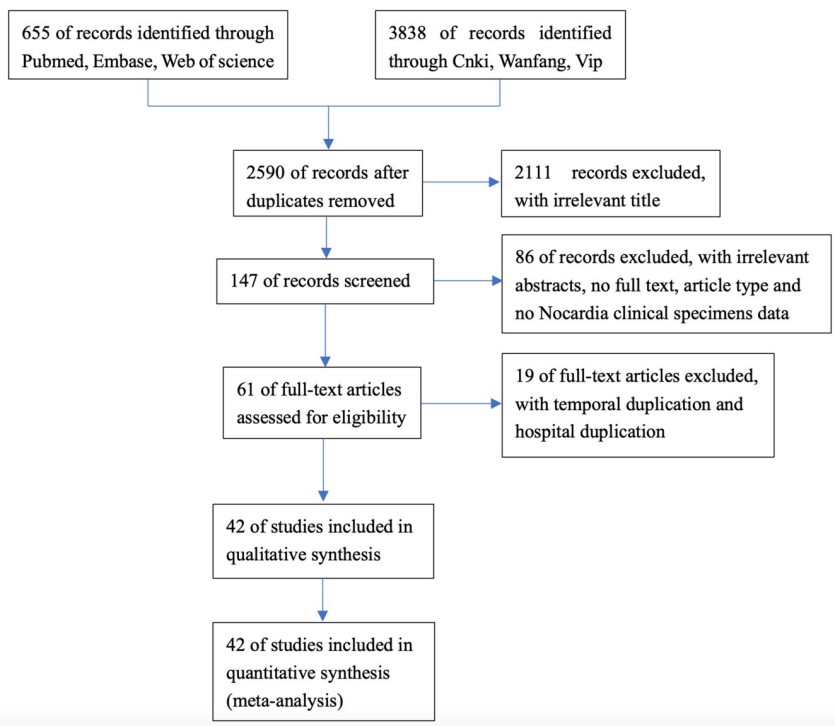

**Fig 1. Flow diagram of study identification.**

*cyriacigeorgica* (25.3%, 200/791), *N. brasiliensis* (11.8%, 93/791) and *N. otitidiscaviarum* (7.8%, 62/791) (Fig 2). The four most common isolated *Nocardia* accounted for 74.0% of all *Nocardia* species.

Both species composition and the number of *Nocardia* isolates demonstrated marked geographic variability. Among 19 *Nocardia* species, 12 isolated from Northeast, 11 from Southeast, 9 from Northwest, 8 from Southwest and 11 from Central China (Table 3). Besides, the eastern region isolated more *Nocardia* strains than central and western region. The southern region had more isolates than the northern region, while the southeastern region had the most strains in China (Fig 3). The prevalence of different *Nocardia* species was also dramatically varied by geographic areas ($\chi^2$ = 249.690, P<0.001). In Northern China, *N.cyriacigeorgica* constituted 36.1% of all the isolated *Nocardia* and 20.3% in case of Southern China. More *N. brasiliensis* were isolated in Western China than in Eastern China (26.5% vs 8.9%, P < 0.001). In central China, *N. farcinica* consisted of 40.0% of all the isolated *Nocardia*. (Table 3).

The prevalent *Nocardia* species showed regional characteristics. *N. farcinica* and *N. cyriacigeorgica* appears widely distributed and mainly prevalent in the northeast and central areas of China, *N. brasiliensis* mainly prevalent in the southwest, *N. otitidiscaviarum* prefers to be distributed in the east coastal provinces (Fig 3). The forest plots prevalence of *N. farcinica*, *N. cyriacigeorgica*, *N. brasiliensis* and *N. otitidiscaviarum* were shown in S1 Fig.

## Stratified analyses of infection type

Totally, 70.4% (223/317) *Nocardia* were cultured from respiratory tract specimens, 16.4% (52/317) from extra-pulmonary specimens, and 13.3% (42/317) from disseminated infection (Fig 4). *N. farcinica* (76/223, 34.1%), *N. brasiliensis* (24/52, 46.2%) and *N. farcinica* (13/42, 31.0%) were the most frequently species causing pulmonary infection, extra-pulmonary infection and

**Table 1. Characteristics of studies involved in the current systematic review and meta-analysis.**

| Authors | Time of study | Publication date | Province | Methods (% of base pair similarities) | *Nocardia* isolation |
|---|---|---|---|---|---|
| Ding et al. [13] | 2017.1–2020.6 | 2022 | Henan | mNGS 16S rRNA sequencing | 14 |
| Zhong et al. [14] | 2011.7–2021.6 | 2022 | Zhejiang | MS | 74 |
| Wei et al. [7] | 2010–2020 | 2021 | Beijing | MLSA | 82 |
| Li et al. [15] | 2013.1–2018.11 | 2021 | Guangdong | 16S rRNA sequencing | 11 |
| Lu et al. [6] | 2007.1–2019.12 | 2020 | Shandong | MS and 16S rRNA sequencing (99.0) | 27 |
| Dong et al. [11] | 2014.1–2017.6 | 2020 | Beijing | MS | 18 |
| Guo et al. [16] | 2010.11–2019.4 | 2020 | Jiangsu | Biochemical tests | 11 |
| Weng et al. [17] | 2014.9–2018.9 | 2020 | Shanghai | NGS | 25 |
| Yi et al. [18] | 2017–2019 | 2019 | Shandong | 16S rRNA sequencing (99.0) MALDI-TOF MS | 19 |
| Huang et al. [10] | 2009–2017 | 2019 | Beijing, Fujian, Guangxi, Hunan, Chongqing, Shandong | MLSA and 16S rRNA sequencing (99.6) and hsp65/secA1 (99.0) /rpoB/gyrB (93.5) sequencing | 53 |
| Zhao et al. [19] | 2010–2015 | 2016 | Beijing | 16S rRNA sequencing (97.0) | 20 |
| Liu et al. [20] | 2005–2012 | 2016 | Gansu, Hunan, Jiangxi | Multilocus PCR sequencing and rpoB and hsp65 PCR-RPA and rpoB, 16S-23S ITS sequencing | 33 |
| Xiao et al. [21] | 2009.1–2015.1 | 2016 | Beijing | MLSA MALDI-TOF MS | 25 |
| Yu et al. [22] | 2011.7–2013.7 | 2014 | Hubei | 16S rRNA and 16S-23S rRNA ITS sequencing (99.0) | 6 |
| Wu et al. [23] | 2017.6–2021.6 | 2022 | Fujian | mNGS | 18 |
| Cheng et al. [24] | 2016.4–2018.12 | 2022 | Hainan | 16S rRNA sequencing (99.6) and hsp65/secA1 (99.0) /gyrB (93.5) sequencing | 10 |
| Gao et al. [25] | 2012–2021 | 2022 | Henan | mNGS | 34 |
| Pan et al. [26] | 2019.1–2020.12 | 2022 | Fujian | 16S rRNA sequencing (99.0) MALDI-TOF MS | 10 |
| Lin et al. [27] | 2015.1–2020.12 | 2021 | Fujian | MS | 10 |
| Cai et al. [28] | 2016.1–2020.11 | 2021 | Sichuan | MS | 13 |
| Liao et al. [29] | 2016.5–2020.10 | 2021 | Hunan | NGS andMS | 24 |
| Xie et al. [30] | 2013.1–2019.7 | 2021 | Shanghai | NGS | 44 |
| Mao et al. [31] | 2002–2019 | 2020 | Fujian | MS | 25 |
| Yang et al. [32] | 2015–2019 | 2020 | Shanxi | 16S rRNA sequencing (99.0) | 15 |
| Zhao et al. [33] | 2013.9–2018.9 | 2020 | Tianjin | MS | 12 |
| Ma et al. [34] | 1976–1998 | 2000 | Beijing | - | 10 |
| Ye et al. [35] | 2012.1–2014.4 | 2014 | Guangdong | - | 7 |

*(Continued)*

**Table 1.** (Continued)

| Authors | Time of study | Publication date | Province | Methods (% of base pair similarities) | *Nocardia* isolation |
|---|---|---|---|---|---|
| Zhang et al. [36] | 2016.7–2017.2 | 2018 | Hunan | 16S rRNA sequencing | 9 |
| Chen et al. [37] | 2016–2019 | 2020 | Hebei | 16S rRNA/secA1 sequencing (99.0) and MALDI-TOF MS | 94 |
| Wang et al. [38] | 2007.1–2014.12 | 2015 | Shandong | 16S rRNA sequencing (99.9) | 8 |
| Chen et al. [39] | 2016.1–2019.9 | 2020 | Sichuan | 16S rRNA sequencing (99) and MALDI-TOF MS | 19 |
| Li et al. [40] | 2012.1–2020.2 | 2020 | Guangxi | NGS and MS | 19 |
| Cheng et al. [41] | 2012.1–2016.6 | 2017 | Ningxia | 16S rRNA sequencing | 20 |
| Gao et al. [42] | 2006.2–2012.9 | 2013 | Anhui | Biochemical tests | 23 |
| Chen et al. [43] | 2005.1–2011.3 | 2012 | Guangxi | Biochemical tests | 13 |
| He et al. [44] | 1990–2009 | 2011 | Henan | - | 16 |
| Lai et al. [45] | 2002.1–2007.1 | 2008 | Zhejiang | - | 5 |
| Nong et al. [46] | 2010–2018 | 2020 | Guangxi | Biochemical tests | 36 |
| Zheng et al. [47] | 2008.1–2019.10 | 2020 | Hunan | MS | 55 |
| Li et al. [48] | 2012.2–2021.5 | 2022 | Sichuan | MS | 21 |
| Chao et al. [49] | 2005.1–2019.12 | 2020 | Qinghai | 16S rRNA (99.0) and rpoB (97.0) sequencing and MALDI-TOF MS | 13 |
| Sun et al. [50] | 2017.1–2021.1 | 2021 | Hebei | MS | 7 |

MS: Mass Spectrometry

mNGS: Metagenomics Next Generation Sequencing

MLSA: Multilocus sequence analysis (gyrB, 16S rRNA, sec A1, rpoB, hsp65)

MALDI-TOF MS: Matrix-assisted laser desorption/ionization-time of flight mass spectrometry

disseminated infection, respectively (Fig 4). The skin and soft tissue wounds (41/52, 78.9%), including feet infection, are the main sources of extra-pulmonary infection. *N. brasiliensis* (24/41, 58.5%) and *N. terpenica* (6/41, 14.6%) are the most common species causing cutaneous nocardiosis.

## Effects of cultural method on *Nocardia* isolation

Among the 42 articles included in this study, 16 of them reported the cultural medium used for *Nocardia* isolation. The isolation rate of *N. farcinica* was 80.3% (53/66) by MGIT 960 mycobacterium liquid medium and 22.8% (71/311) using routine columbia blood plate. There was a significant difference ($\chi^2 = 81.478$, $P < 0.001$).

## Antimicrobial susceptibility profiles

There were 198 isolates with available antibiotic susceptibility data, which performed using the standard Broth microdilution method. The proportion of susceptible isolates as follows:

**Table 2. Species distribution among the *Nocardia* isolates from China.**

| *Nocardia* species | No. of study | N(%) | Prevalence of *Nocardia* (95% CI) | Heterogeneity | Heterogeneity P-value | Egger's test t | Egger's test P-value |
|---|---|---|---|---|---|---|---|
| *N. farcinica* | 32 | 230 (29.1) | 0.23 [0.17,0.30] | 85.6 | <0.001 | 2.19 | 0.037 |
| *N. cyriacigeorgica* | 24 | 200 (25.3) | 0.24 [0.17,0.31] | 85.4 | <0.001 | 4.14 | 0.000 |
| *N. brasiliensis* | 32 | 93 (11.8) | 0.11 [0.08,0.14] | 56.4 | <0.001 | 11.22 | 0.000 |
| *N. otitidiscaviarum* | 22 | 62 (7.8) | 0.07 [0.05,0.10] | 20.4 | 0.193 | 3.64 | 0.002 |
| *Nocardia.spp* | 22 | 104 (13.2) | 0.24 [0.15,0.32] | 88.0 | 0.340 | 5.05 | 0.000 |
| *N. abscessus* | 8 | 26 (3.3) | 0.07 [0.04,0.09] | 0.0 | 0.818 | 1.83 | 0.118 |
| *N. terpenica* | 5 | 17 (2.2) | 0.09 [0.02,0.16] | 57.7 | 0.051 | 2.90 | 0.063 |
| *N. nova* | 10 | 13 (1.6) | 0.02 [0.01,0.03] | 0.0 | 0.750 | 8.39 | 0.000 |
| *N. asiatica* | 6 | 13 (1.6) | 0.04 [0.02,0.06] | 0.0 | 0.958 | 1.34 | 0.251 |
| *N. wallacei* | 3 | 8 (1.0) | 0.03 [0.01,0.06] | 0.0 | 0.425 | 2.48 | 0.244 |
| *N. puris* | 4 | 7 (0.9) | 0.03 [0.00,0.05] | 4.0 | 0.373 | 4.89 | 0.039 |
| *N. beijingensis* | 4 | 5 (0.6) | 0.02 [-0.01,0.04] | 10.9 | 0.338 | 5.70 | 0.029 |
| *N. transvalensis* | 3 | 3 (0.4) | 0.01 [-0.00,0.03] | 0.0 | 0.895 | - | - |
| *N. pseudobrasilliensis* | 2 | 2 (0.3) | 0.02 [-0.01,0.06] | 0.0 | 0.611 | - | - |
| *N. blacklockiae* | 1 | 2 (0.3) | 0.04 [-0.01,0.09] | - | - | - | - |
| *N. araoensis* | 1 | 2 (0.3) | 0.20 [-0.05,0.45] | - | - | - | - |
| *N. aobensis* | 1 | 1 (0.1) | 0.01 [-0.01,0.04] | - | - | - | - |
| *N. vulneris* | 1 | 1 (0.1) | 0.10 [-0.09,0.29] | - | - | - | - |
| *N. africana* | 1 | 1 (0.1) | 0.02 [-0.02,0.05] | - | - | - | - |
| *N. concava* | 1 | 1 (0.1) | 0.08 [-0.07,0.22] | - | - | - | - |

linezolid 99.5% (197/198), amikacin 96.0% (190/198), trimethoprim-sulfamethoxazole 92.9% (184/198), imipenem 64.7% (128/198), tobramycin 63.1% (99/157), gentamicin 62.0% (31/50), minocycline 50.7% (75/148), ceftriaxone 45.7% (80/175) (Table 4). Susceptibility varied by species of *Nocardia*. While susceptibility for minocycline for *N. otitidiscaviarum* and *N. brasiliensis* was 86.4% and 88.9%, respectively; susceptibility for *N. farcinica* and *N. cyriacigeorgica* were lower at 21.2% and 38.3%, respectively. *N. cyriacigeorgica* (90.1%) showed relatively high susceptible rates to imipenem, while *N. otitidiscaviarum* (7.7%), *N. brasiliensis* (33.3%) and *N. farcinica* (45.5%) showed relatively low susceptible rates to imipenem. Most species showed low susceptibility rates to moxifloxacin; however, 79.3% (23/29) *N. farcinica* and 66.7% (4/6) *N. brasiliensis* isolates were susceptible to moxifloxacin, respectively (Table 4).

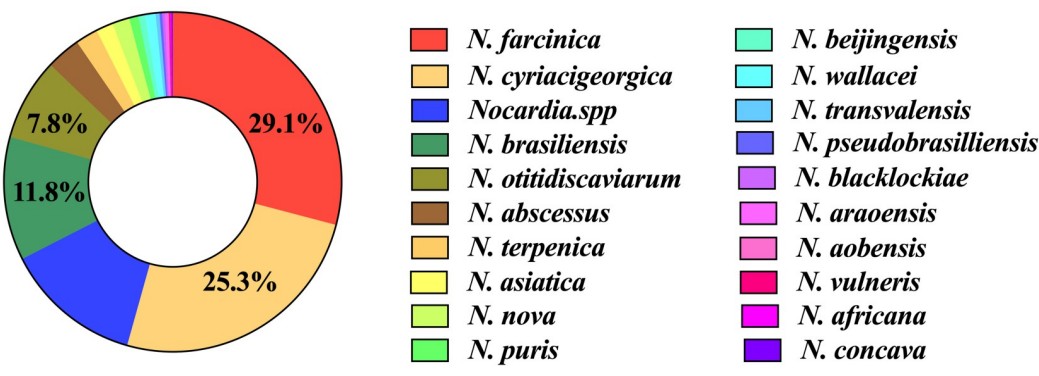

**Fig 2. The species distribution of 791 *Nocardia* isolates.**

**Table 3. Species distribution of the *Nocardia* isolates among five geographic areas in China.**

| *Nocardia* species | Northeast | Southeast | Northwest | Southwest | Central |
|---|---|---|---|---|---|
| *N. farcinica* | 79 (31.1) | 39 (20.1) | 7 (15.6) | 9 (10.3) | 50 (40.0) |
| *N. brasiliensis* | 15 (5.9) | 25 (12.9) | 7 (15.6) | 28 (32.2) | 16 (12.8) |
| *N. cyriacigeorgica* | 92 (36.2) | 51 (26.3) | 16 (35.6) | 6 (6.9) | 24 (19.2) |
| *N. otitidiscaviarum* | 19 (7.5) | 20 (10.3) | 6 (13.3) | 5 (5.8) | 8 (6.4) |
| *N. nova* | 5 (2.0) | 1 (0.5) | 0 (0.0) | 1 (1.2) | 3 (2.4) |
| *N. abscessus* | 14 (5.5) | 1 (0.5) | 3 (6.7) | 2 (2.3) | 1 (0.8) |
| *N. asiatica* | 8 (3.2) | 1 (0.5) | 1 (2.2) | 1 (1.2) | 0 (0.0) |
| *N. terpenica* | 0 (0.0) | 7 (3.6) | 0 (0.0) | 1 (1.2) | 1 (0.8) |
| *N. puris* | 4 (1.6) | 0 (0.0) | 1 (2.2) | 0 (0.0) | 2 (1.6) |
| *N. beijingensis* | 3 (1.2) | 0 (0.0) | 1 (2.2) | 0 (0.0) | 0 (0.0) |
| *N. wallacei* | 6 (2.4) | 2 (1.0) | 0 (0.0) | 0 (0.0) | 0 (0.0) |
| *N. transvalensis* | 1 (0.4) | 0 (0.0) | 0 (0.0) | 0 (0.0) | 1 (0.8) |
| *Nocardia.spp* | 7 (2.8) | 44 (22.7) | 2 (4.4) | 34 (39.1) | 17 (13.6) |
| *N. pseudobrasilliensis* | 0 (0.0) | 0 (0.0) | 0 (0.0) | 0 (0.0) | 1 (0.8) |
| *N. blacklockiae* | 0 (0.0) | 0 (0.0) | 0 (0.0) | 0 (0.0) | 0 (0.0) |
| *N. araoensis* | 0 (0.0) | 2 (1.0) | 0 (0.0) | 0 (0.0) | 0 (0.0) |
| *N. aobensis* | 1 (0.4) | 0 (0.0) | 0 (0.0) | 0 (0.0) | 0 (0.0) |
| *N. vulneris* | 0 (0.0) | 1 (0.5) | 0 (0.0) | 0 (0.0) | 0 (0.0) |
| *N. africana* | 0 (0.0) | 0 (0.0) | 0 (0.0) | 0 (0.0) | 1 (0.8) |
| *N. concava* | 0 (0.0) | 0 (0.0) | 1 (2.2) | 0 (0.0) | 0 (0.0) |

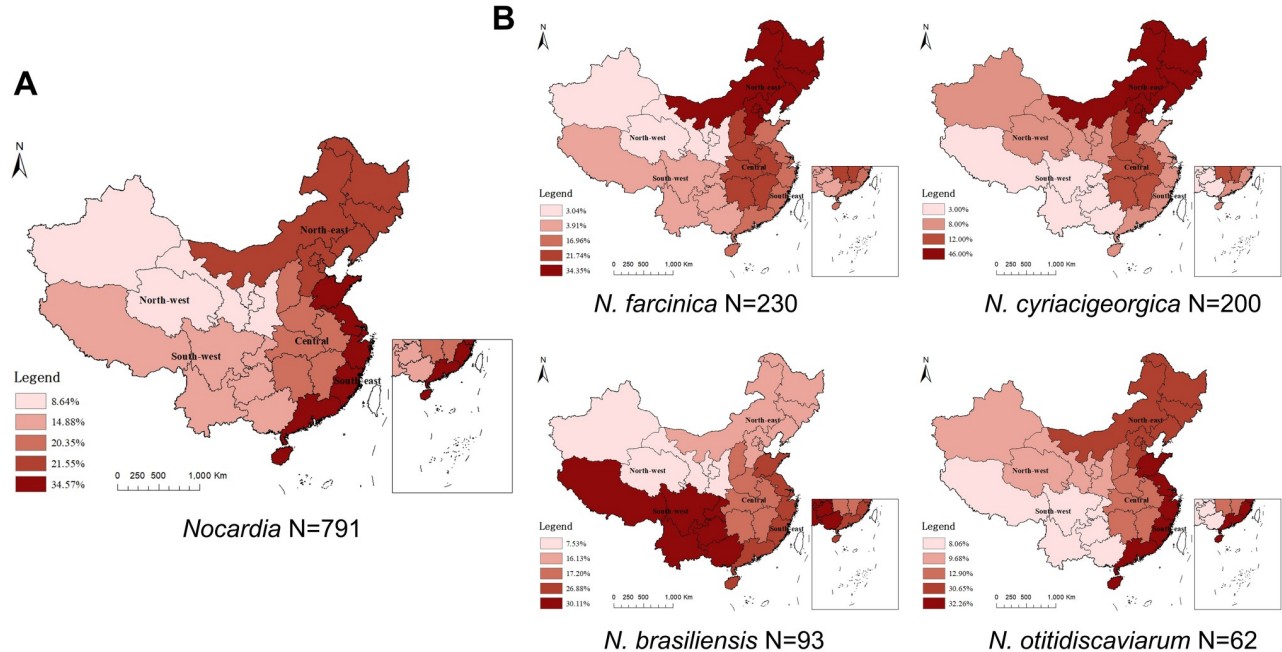

**Fig 3. Distribution of *Nocardia* isolates in five geographic areas of China.** Distribution of Nocardia isolates in five geographic areas of China. (A) Geographical locations of the total Nocardia isolates, (B) Distribution of N. farcinica, N. cyriacigeorgica, N. brasiliensis and N. otitidiscaviarum. The color-highlighted areas represent those where Nocardia isolates were collected, with the proportion of strains provided in parentheses. (Source of data is the latest version of National Basic Geographic Information System, review number GS(2020)4619).

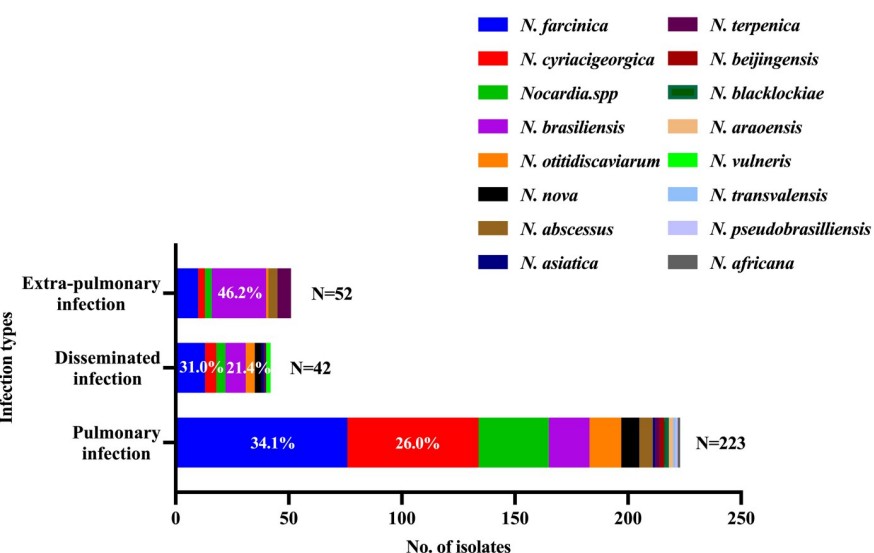

**Fig 4. *Nocardia* species distribution grouped by infection types.**

## Discussion

*Nocardia* are ubiquitous in the environment and could lead to life-threatening infection. With gradually increased incidence, nocardiosis has become a noticeable health threat in China. *Nocardia* infectons are prone to be misdiagnosed as TB or multidrug-resistant TB [3], when only acid-fast staining and mycobacterial culture are used for the diagnosis of TB [51]. Although increasing attention towards *Nocardia* infection, there is not an organized

**Table 4. Antimicrobial susceptibility results of *Nocardia* isolates in China.**

| Antibiotics | Total | *N. farcinica* | *N. cyriacigeorgica* | *N. otitidiscaviarum* | *N. abscessus* | *N. brasiliensis* | *N. terpenica* |
|---|---|---|---|---|---|---|---|
| Trimethoprim-sulfamethoxazole | 92.9 (184/198) | 83.6 (46/55) | 98.6 (70/71) | 100.0 (26/26) | 100.0 (11/11) | 100.0 (9/9) | 75.0 (6/8) |
| Linezolid | 99.5 (197/198) | 100.0 (55/55) | 98.6 (70/71) | 100.0 (26/26) | 100.0 (11/11) | 100.0 (9/9) | 100.0 (8/8) |
| Amikacin | 96.0 (190/198) | 98.2 (54/55) | 100.0 (71/71) | 100.0 (26/26) | 100.0 (11/11) | 100.0 (9/9) | 87.5 (7/8) |
| Imipenem | 64.7 (128/198) | 45.5 (25/55) | 90.1 (64/71) | 7.7 (2/26) | 100.0 (11/11) | 33.3 (3/9) | 100.0 (8/8) |
| Tobramycin | 63.1 (99/157) | 2.4 (1/42) | 91.7 (55/60) | 90.9 (20/22) | 100.0 (6/6) | 100.0 (9/9) | - |
| Gentamicin | 62.0 (31/50) | 13.6 (3/22) | 100.0 (11/11) | 100.0 (4/4) | 100.0 (5/5) | - | 100.0 (8/8) |
| Minocycline | 50.7 (75/148) | 21.2 (7/33) | 38.3 (23/60) | 86.4 (19/22) | 100.0 (6/6) | 88.9 (8/9) | - |
| Ceftriaxone | 45.7 (80/175) | 14.3 (6/42) | 63.5 (40/63) | 0.0 (0/25) | 100.0 (11/11) | 12.5 (1/8) | 75.0 (6/8) |
| Ciprofloxacin | 42.4 (84/198) | 69.1 (38/55) | 22.5 (16/71) | 53.9 (14/26) | 18.2 (2/11) | 11.1 (1/9) | 100.0 (8/8) |
| Cefepime | 41.7 (73/175) | 9.5 (4/42) | 47.6 (30/63) | 16.0 (4/25) | 100.0 (11/11) | 12.5 (1/8) | 100.0 (8/8) |
| Cefotaxime | 43.8 (28/64) | 23.1 (6/26) | 73.7 (14/19) | 0.0 (0/25) | 0.0 (0/5) | 0.0 (0/1) | 100.0 (8/8) |
| Cefoxitin | 3.6 (1/28) | 0.0 (0/17) | 0.0 (0/6) | 0.0 (0/2) | - | 0.0 (0/2) | - |
| Cefatadine | 13.9 (5/36) | 0.0 (0/10) | 0.0 (0/7) | 0.0 (0/11) | 100.0 (1/1) | 0.0 (0/3) | - |
| Doxycycline | 18.6 (18/97) | 3.6 (1/28) | 100.0 (6/6) | 33.3 (3/9) | 100.0 (5/5) | 0.0 (0/3) | - |
| Tegacyclin | 94.7 (18/19) | 87.5 (7/8) | - | 100.0 (2/2) | - | 100 (2/2) | - |
| Moxifloxacin | 30.7 (38/124) | 79.3 (23/29) | 4.4 (2/46) | 35.0 (7/20) | 16.7 (1/6) | 66.7 (4/6) | - |
| Amoxicillin Clavulanic acid | 40.0 (70/175) | 81.0 (34/42) | 17.5 (11/63) | 0.0 (0/25) | 100.0 (11/11) | 75.0 (6/8) | 12.5 (1/8) |
| Clarithromycin | 9.0 (12/134) | 0.0 (0/29) | 11.5 (6/52) | 0.0 (0/21) | 0.0 (0/6) | 0.0 (0/8) | - |

monitoring system for *Nocardia* spp. An epidemiological analysis of *Nocardia* spp. mainly depends on meta-analysis or systematic review. This study will build a nationwide overview of species distribution and antimicrobial resistance profiles of *Nocardia* species in China.

The species distribution of *Nocardia* has unique characteristics worldwide. Our meta-analysis showed that the most common species was *N. farcinica* (29.1%, 230/791), followed by *N. cyriacigeorgica* (25.3%, 200/791), *N. brasiliensis* (11.8%, 93/791) and *N. otitidiscaviarum* (7.8%, 62/791) in China. Internationally, *N. farcinica* was the most common species in South Africa (20.5%) [52], Belgium (44%) [53], and France (20.2%) [54], whilst *N. cyriacigeorgica* was the most common species in Spain (25.3%) [55] and Iran (31.0%) [56], and *N. nova* was the most common species in the United States (21.6–28%) [57–59] and Australia (29–35.5%) [7,59]. Furthermore, the species composition from different provinces and climates of China demonstrated marked variations. *N. farcinica* and *N. cyriacigeorgica* are widely distributed. *N. brasiliensis* mainly prevalent in the south, which belongs to the subtropical monsoon climate. *N. otitidiscaviarum* mainly distributed in the eastern coastal provinces of China and it is reasonable to assume that it is determined by the sea. Considering the large size of Chinese territories and different climate conditions, it is important to build a knowledgebase of local prevalence of *Nocardia* species.

Different species may cause different types of infection. As expected, the most common source for positive *Nocardia* cultures was respiratory tract specimens in China. Although the frequent association of *N. farcinica* with brain abscesses, bacteremia, skin and subcutaneous infection has been reported [55,60], in our study, in 223 pulmonary infection cases, 76 (34.1%) were *N. farcinica*. Moreover, *N. brasiliensis* was related predominately to cutaneous nocardiosis [9]. In our study, of 51 *N. brasiliensis* isolates, 24 (47.1%) were recovered from the skin and soft tissue infections.

*Nocardia* strains are frequently isolated during culture for mycobacteria in high TB burden settings, however, procedures used for decontamination of sputum specimens may be deleterious to *Nocardia* isolates [61]. Another factor that may limit *Nocardia* recovery is that egg-base L-J media is not an optimal choice for *Nocardia* isolation [11,62]. Furthermore, there are 5 antibiotics in MGIT 960 mycobacterium liquid media, which maybe unfavorable to some *Nocardia* species. Although *Nocardia* seem to grow well on blood agar and fungal media, some strains may be inhibited by the gentamicin present in inhibitory mold agar [62]. The isolated *Nocardia* species varied according to the culture methods used.

This systemic study explored the association of antimicrobial susceptibility profiles and *Nocardia* species to reach a guideline for the nocardiosis treatment in China. Trimethoprim-sulfamethoxazole constitutes the keystone of nocardiosis treatment [57], while the resistance rates for *Nocardia* varied among different regions worldwide [57,63]. Only 7.1% (14/198) of all *Nocardia* isolates were resistant to trimethoprim-sulfamethoxazole in our study, majority of which were *N. farcinica* (83.6%, 46/55). Our results indicate that trimethoprim-sulfamethoxazole is frequently activite against *Nocardia* species isolated in China, and could be used as the primary agent to treat nocardiosis, even without antibiotic susceptibility results. Linezolid and amikacin are also effective drugs for most *Nocardia* species, with 99.5% (197/198) and 96.0% (190/198) activity against clinical isolates, respectively. Linezolid and amikacin could be potentially used for empiric treatment of nocardiosis in China. Imipenem showed good activity for *N. cyriacigeorgica* with susceptibility rate as 90.1% (64/71) in the current study, however, *N. farcinica* (45.5%, 25/55) showed relatively low susceptible rates to imipenem. The remaining antimicrobials showed low activity against *Nocardia* isolates, and the susceptibility profiles were highly variable between different species. Thus, it is imperative to identify *Nocardia* isolates to the species level and an antimicrobial susceptibility test should be conducted during clinical practice to properly treat nocardiosis. Owing to the absence or delay of species

identification and drug susceptibility outcomes, nocardiosis patients may be treated empirical. It is helpful to refer to the local drug resistance prevalence data for the initial drug choice to treat *Nocardia* infection. Our results will begin to better understand *Nocardia* species distributed in China and for the choice of empirical therapy.

The present study is the first meta-analysis of *Nocardia* species distribution and antimicrobial susceptibility patterns in China, but its limitations should also be noted. *Nocardia* infection is not required to be reported to public health authorities, hence its precise prevalence in China is not available. There were few studies on the incidence of nocardiosis in China. Most of published data focus on clinical manifestations, species distribution, and antimicrobial susceptibility of *Nocardia* species. Our study could not show the overall prevalence rate of *Nocardia* infection in China.

## Conclusion

Pulmonary nocardiosis is the most common type of infection. *N. farcinica* and *N. cyriacigeorgica* are the most frequently isolated species, which are widely distributed in China. Trimethoprim-sulfamethoxazole can still be the preferred agent for initial *Nocardia* infection therapy due to the low resistance rate, and linezolid and amikacin could be an alternative to treat nocardiosis or a choice in a combination regimen.

## Supporting information

**S1 Table. PRISMA 2020 checklist.**
(DOCX)

**S1 Fig. Forest plots of proportion of *N. farcinica*, *N. cyriacigeorgica*, *N. brasiliensis* and *N. otitidiscaviarum* in China.**
(TIF)

## Author Contributions

**Conceptualization:** Chaohong Wang, Guirong Wang, Junhua Pan.

**Data curation:** Chaohong Wang, Qing Sun, Jun Yan, Xinlei Liao, Sibo Long, Maike Zheng, Yun Zhang, Xinting Yang, Guangli Shi, Yan Zhao.

**Formal analysis:** Chaohong Wang.

**Funding acquisition:** Guirong Wang, Junhua Pan.

**Investigation:** Chaohong Wang, Qing Sun, Jun Yan.

**Methodology:** Chaohong Wang.

**Project administration:** Guirong Wang, Junhua Pan.

**Resources:** Guirong Wang.

**Software:** Chaohong Wang, Maike Zheng.

**Supervision:** Chaohong Wang.

**Validation:** Qing Sun, Jun Yan.

**Visualization:** Qing Sun, Jun Yan.

**Writing – original draft:** Chaohong Wang.

**Writing – review & editing:** Chaohong Wang, Guirong Wang.

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
