## [Decision Letter · Decision Letter 0]

31 Mar 2023

Dear Prof Wang,

Thank you very much for submitting your manuscript "The species distribution and antimicrobial resistance profiles of Nocardia species in China: a systematic review and meta-analysis" for consideration at PLOS Neglected Tropical Diseases. As with all papers reviewed by the journal, your manuscript was reviewed by members of the editorial board and by several independent reviewers. In light of the reviews (below this email), we would like to invite the resubmission of a significantly-revised version that takes into account the reviewers' comments. 

I want to apologise for the extreme delay you have experienced in waiting for the reviews on your manuscript. It was uncharacteristically difficult to secure three reviews in this instance, but as you can see that has indeed now occured. The reviews largely agree that the manuscript was well written and offers useful insights that would benefit the research community. Each reviewer has identified numerous edits they would like to see in a revised manuscript and I would concur with these suggestions and recommend you address each of them in turn. I look forward to receiving a revised manuscript from you and, once again, my apologies for the delay.

We cannot make any decision about publication until we have seen the revised manuscript and your response to the reviewers' comments. Your revised manuscript is also likely to be sent to reviewers for further evaluation.

Sincerely,

Andrew C Singer, Ph.D.

Academic Editor

Elsio Wunder Jr

Section Editor

I want to apologise for the extreme delay you have experienced in waiting for the reviews on your manuscript. It was uncharacteristically difficult to secure three reviews in this instance, but as you can see that has indeed now occured. The reviews largely agree that the manuscript was well written and offers useful insights that would benefit the research community. Each reviewer has identified numerous edits they would like to see in a revised manuscript and I would concur with these suggestions and recommend you address each of them in turn. I look forward to receiving a revised manuscript from you and, once again, my apologies for the delay.

Reviewer's Responses to Questions

**Summary and General Comments**

Reviewer #1: The authors conducted an interesting systemic review and meta-analysis to describe the species distribution and antimicrobial resistance profiles of Nocardia species in China. But this article did not fulfill the spirit of meta-analysis. Meta-analysis is a research process used to systematically synthesise or merge the findings of single, independent studies, using statistical methods to calculate an overall or ‘absolute’ effect. Meta-analysis does not simply pool data from smaller studies to achieve a larger sample size. Analysts use well recognised, systematic methods to account for differences in sample size, variability (heterogeneity) in study approach and findings (treatment effects) and test how sensitive their results are to their own systematic review protocol (study selection and statistical analysis). (https://ebn.bmj.com/content/16/1/3) Second, this study did not bring new novelty or new findings to help clinicians.

There were some issues that needed to be addressed: 

1. The manuscript should be checked again for spelling/grammatical mistakes and correct punctuation, e.g. N. farcinica, N. cyriacigeorgica (a space between genus and species), genus/species in italic, gradual increasing (gradually increasing?), Nocardiosis resemble tuberculosis (Nocardiosis resembles tuberculosis?), cinicains (clinicians?), involving 1008 clinical Nocardia isolated (Nocardia isolates?), etc. Make sure that the numbers were correct in regards to those in tables (66.67 %

(28/28) N. brasiliensis isolates were susceptible to moxifloxacin??)

2. A paragraph about the method of paper quality assessment should be added (PRISMA 2020 checklist/QUADAS-2). It should also be included in the result section or supplementary material.

3. In table 1, the method column and province column should be filled with correct method (e.g. article 4 used PCR for species identification) or cities (multiple provinces should also be stated), not just filled with “-”.

4. The distribution of extra-pulmonary specimens should be presented if possible. Different species may cause different types of infection.

5. It may be better to change the sentence “linezolid could be an alternative. Amikacin can be the choice in a combination regimen” to “linezolid and amikacin could be an alternative to treat nocardiosis or a choice in a combination regimen”.

Reviewer #2: Comments on "The species distribution and antimicrobial resistance profiles of Nocardia species in China: a systematic review and meta-analysis"

This meta-analysis aims at describing molecular epidemiology at the species level of Nocardia in China. They also analysed antimicrobial resistance profiles.

Overall, the paper is clearly written and the topic is of interest.

However, several major comments need to be taken into account.

Major comments

- Throughout manuscript, the authors report a high proportion of N. asteroides.

It is well documented now that N. asteroides is an exceptional human pathogen. Most of reported N. asteroides have been misidentified. It is therefore very likely that the different proportions of N. asteroides (depending on the cited paper) is rather caused by different proportions of misidentification (depending on the method that is used). As a conclusion, all N. asteroides should be reported as Nocardia spp. and proportions of other species should be recalculated.

This point should be included in the Methods, results and in the discussion section + figures and tables

-The authors claim in the Introduction and Discussion sections that "species distribution and antimicrobial resistance profiles remain unavailable for China". As the authors performed a meta-analysis of previously published studies, this statement is, by definition, wrong. Otherwise, metanalysis couls not have been performed. These data exist but the aim of this meta analysis was to bring them together and build a nationwide overview. This should be stated

- Important bibliographical data are not precise enough, and even wrong.

Introduction: no data support the statement that the incidence of nocardiosis is increasing. The cited paper (REF 2) does not even mention this question.

Introduction and discussion: sentences regarding tuberculosis do not add anything to the paper. This can also be misleading as tuberculosis and nocardiosis are many differences. I would remove these sentences that do not belong to this story.

- Important methodological aspects are lacking

Table 1: the authors mention "PCR" as a "method". What does it mean? Was it the technic for species identification. If so, was it amplification and sequencing or detection? If it was amplification, what was the target gene? What were the interpretation rules (% of base pair similarities)? This should be detailed in Table 1.

In the same line, methods for antibiotic susceptilibility testing are not detailed. In this context, it is not possible to compare % of susceptibility.

Xas it broth microdilution, E-tests, antibiotic disk diffusion on agar plates?

Minor comments.

- Throughout manuscript, please add a space after the dot for bacterial names: "N. farcinica" rather that "N.farcinica".

- For percentage, I suggest limiting to one value after the dot: 70.3% rather than 70.35%

- page 14, "causing" rather that "caused"

- Discussion: page 19, the sentence regarding trimethoprim/sulfamethoxazole is not accurate. The authors assessed the % of suspectibility but not the "activity". They cannot state that "trimethoprim/sulfamethoxazole still has high-level activity."... They can only state that "trimethoprim/sulfamethoxazole is frequently activite".

Reviewer #3: This manuscript is well written and presents all data in a clear, organized fashion. The table presenting the references for the systematic review displays the method of species ID found in each reference and is very valuable/helpful to the reader. 

Please insert line numbers to aid in directing comments to specific text. 

In the discussion on the geographic distribution of reported Nocardia species in China, please add a little information about the climate and general ecosystem in each of the regions that have been defined. 

In the section on cultural method on Nocardia isolation, it is reported that 80.33% of Nocardia farcinica were isolated by MGIT 960 Myco medium. Were NTM or TB co-infections reported in any of those cases?

In the discussion (and throughout the manuscript) Nocardia asteroides is noted as one of the prevalent species in the southern regions of China. However, reports of Nocardia asteroides sensu stricto are quite rare and there is much confusion over the accuracy of any species reported as Nocardia asteroides. Is there assurance that these isolates were in fact accurately identified. Or could this be a misrepresentation due to lack of accurate ID?

Please give a reference for the following statement in the discussion, “Another factor that that may limit Nocardia recovery is that egg-based LJ media is not an optimal choice for Nocardia isolation.”

PLOS authors have the option to publish the peer review history of their article (what does this mean?). If published, this will include your full peer review and any attached files.

Reviewer #1: No

Reviewer #2: Yes: David Lebeaux

Reviewer #3: No
---

## [Editor Report · Decision Letter 1]

5 Jun 2023

Dear Prof Wang,

We are pleased to inform you that your manuscript 'The species distribution and antimicrobial resistance profiles of Nocardia species in China: a systematic review and meta-analysis' has been provisionally accepted for publication in PLOS Neglected Tropical Diseases.

Best regards,

Andrew C Singer, Ph.D.

Academic Editor

Elsio Wunder Jr

Section Editor

Thank you for your attention to the reviewer feedback. I can confirm that your manuscript has been accepted for publication.

---

## [Editor Report · Acceptance letter]

5 Jul 2023

Dear Prof Wang,

We are delighted to inform you that your manuscript, "The species distribution and antimicrobial resistance profiles of Nocardia species in China: a systematic review and meta-analysis," has been formally accepted for publication in PLOS Neglected Tropical Diseases.

Best regards,

Shaden Kamhawi

co-Editor-in-Chief

Paul Brindley

co-Editor-in-Chief
